# Genome-wide association study in frontal fibrosing alopecia identifies four susceptibility loci including *HLA-B\*07:02*

Christos Tziotzios (ORCID) et al.[#]

Frontal fibrosing alopecia (FFA) is a recently described inflammatory and scarring type of hair loss affecting almost exclusively women. Despite a dramatic recent increase in incidence the aetiopathogenesis of FFA remains unknown. We undertake genome-wide association studies in females from a UK cohort, comprising 844 cases and 3,760 controls, a Spanish cohort of 172 cases and 385 controls, and perform statistical meta-analysis. We observe genome-wide significant association with FFA at four genomic loci: 2p22.2, 6p21.1, 8q24.22 and 15q2.1. Within the 6p21.1 locus, fine-mapping indicates that the association is driven by the *HLA-B\*07:02* allele. At 2p22.1, we implicate a putative causal missense variant in *CYP1B1*, encoding the homonymous xenobiotic- and hormone-processing enzyme. Transcriptomic analysis of affected scalp tissue highlights overrepresentation of transcripts encoding components of innate and adaptive immune response pathways. These findings provide insight into disease pathogenesis and characterise FFA as a genetically predisposed immuno-inflammatory disorder driven by *HLA-B\*07:02*.

Frontal fibrosing alopecia (FFA) is a recently reported lichenoid and scarring inflammatory skin disorder associated with widespread cutaneous inflammation and irreversible hair loss, which occurs predominantly in women of postmenopausal age (Fig. 1)[1,2]. Since FFA was first identified by Kossard in 1994, there has been rapid increase in reported incidence culminating in intense clinical and public interest in the condition. FFA is often referred to as a dermatological epidemic with possible environmental trigger(s) implicated[3]. Nevertheless, the pathogenesis of FFA also includes a genetic component, as evidenced by frequent familial segregation[4–11].

FFA is considered to be a clinical sub-variant of lichen planus, a more common inflammatory skin condition of unresolved aetiology, while also representing a variant of the prototypic primary lymphocytic cicatricial (or scarring) alopecia lichen planopilaris (LPP). A key molecular event in the pathology of scarring hair loss has been postulated to be the immune privilege collapse at the level of the immunologically shielded hair follicle bulge, which is home to epithelial hair follicle stem cells (eHFSC): T-cell mediated inflammatory presence culminates in stem cell apoptosis and irreversible alopecia[12]. Dissection of the genetic basis of FFA and its interplay with environmental risk factors, therefore, could provide insight into the molecular profile of lichenoid inflammation, scarring and mechanisms of immune privilege collapse. Furthermore, the identification of environmental triggers that interact with FFA genetic susceptibility loci could ultimately contribute to disease prevention by avoidance of exposures in genetically predisposed individuals (Supplementary Note 1).

To date there have been no systematic investigations into the molecular genetic basis of FFA or any other lichenoid inflammatory disorder. We hypothesise that common genetic variation contributes to FFA susceptibility and undertake a genome-wide association study and meta-analysis of two independent European cohorts of females with FFA and controls and investigate transcriptomic and metabolomic involvement in the disease.

## Results

**Genome-wide association study**. We undertook a genome-wide association analysis across 8,405,903 common variants in a UK cohort of 844 FFA female cases and 3760 female controls. Inspection of the quantile-quantile plot indicated adequate control of confounding bias ($\lambda_{GC(MAF>0.05)} = 1.03$; Supplementary Table 2; Supplementary Figure 3). We observed genetic variants with genome-wide significant association ($P < 5.0 \times 10^{-8}$) with FFA at three genomic loci; 6p21.1, 2p22.2 and 15q26.1 (Table 1). We estimate the genome-wide SNP heritability for FFA as 46.66% (SE = 3.00%).

In an attempt to replicate the observed associations at each of the three loci and identify additional FFA susceptibility loci, we performed a genome-wide association study across 7,964,651 common variants in our independent Spanish cohort comprising 172 affected females and 385 controls. We observed allelic associations with FFA at each of the three loci that had been implicated in FFA susceptibility in the UK cohort. The direction and magnitude of the effect of these associations was consistent between the UK and Spanish cohorts (Table 1). To identify further loci harbouring variation contributing to FFA risk we performed a statistical meta-analysis of the association summary statistics from the UK and Spanish cohorts. This revealed a single additional risk locus at 8q24.22, again with a consistent direction and magnitude of effect in both studies (Fig. 2 and Table 1) and a number of loci at which there is suggestive evidence of association ($P < 5 \times 10^{-5}$; Supplementary Table 3).

We sought to further investigate the allelic basis for the observed FFA association at 6p21.1, which is located within the MHC region. We undertook imputation of classical MHC Class I alleles and evaluated the association of each allele with FFA. The strongest evidence of association was observed for the HLA-B allele *HLA-B\*07:02* ($P_{Meta} = 9.44 \times 10^{-117}$, OR = 5.22 (4.53–6.01); Supplementary Table 4), indicating that this is the most likely classical HLA allele to be underpinning the observed SNP associations in this region. Although full characterization of

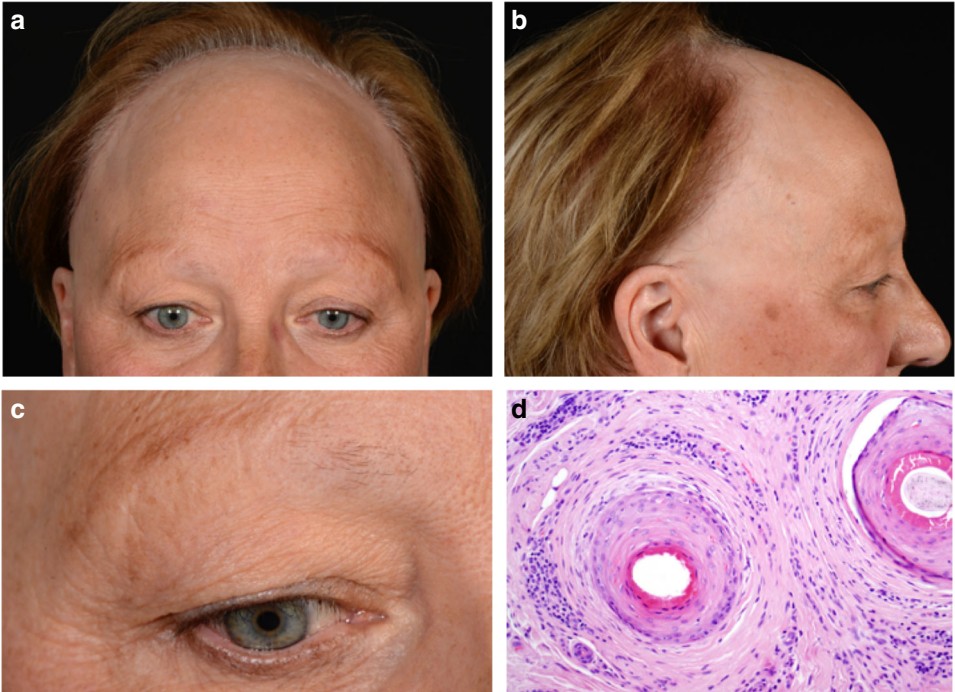

**Fig. 1** Clinical features of frontal fibrosing alopecia. Scalp with frontal hairline recession (**a**) involving the temporal areas bilaterally (**b**), as well as eyebrows (**c**). Histopathology (**d**) shows two hair follicles with focal interface changes, and a moderately dense perifollicular lymphoid cell infiltrate with perifollicular fibrosis, characteristic of FFA (×200)

**Table 1 Genome-wide significant loci for UK, Spain and meta-analysis**

| Locus | Gene | Position (hg19) | SNP ID | RA | PA | RAF Cases | RAF Controls | UK cohort | | Spanish cohort | | Meta-analysis | |
|---|---|---|---|---|---|---|---|---|---|---|---|---|---|
| | | | | | | | | OR (95% CI) | P | OR (95% CI) | P | OR (95% CI) | P |
| 2p22.2 | CYP1B1 | 38,298,139 | rs1800440 | T | C | 0.87 | 0.81 | 1.62 (1.38–1.90) | $5.89 \times 10^{-9}$ | 1.81 (1.28–2.58) | 0.00090 | 1.65 (1.43–1.91) | $2.44 \times 10^{-11}$ |
| 6p21.1 | HLA-B | 31,320,562 | rs2523616 | T | C | 0.47 | 0.19 | 4.69 (4.07–5.40) | $8.52 \times 10^{-101}$ | 4.97 (3.52–7.02) | $8.09 \times 10^{-20}$ | 4.73 (4.15–5.39) | $7.60 \times 10^{-119}$ |
| 8q24.22 | ST3GAL1 | 134,503,229 | rs760327 | G | C | 0.46 | 0.39 | 1.32 (1.18–1.47) | $1.18 \times 10^{-6}$ | 1.50 (1.14–1.97) | 0.00357 | 1.34 (1.21–1.49) | $2.15 \times 10^{-8}$ |
| 15q26.1 | SEMA4B | 90,734,426 | rs34560261 | T | C | 0.22 | 0.17 | 1.52 (1.32–1.76) | $8.47 \times 10^{-9}$ | 1.51 (1.03–2.21) | 0.03257 | 1.52 (1.22–1.74) | $8.12 \times 10^{-10}$ |

Each SNP was tested for association by logistic regression using an additive regression model; total $N = 5161$ biologically independent subjects ($N_{cases} = 1044$ and $N_{controls} = 4145$)

RA risk allele, PA protective allele, RAF risk allele frequency, OR odds ratio, RAF risk allele frequency, CI confidence interval

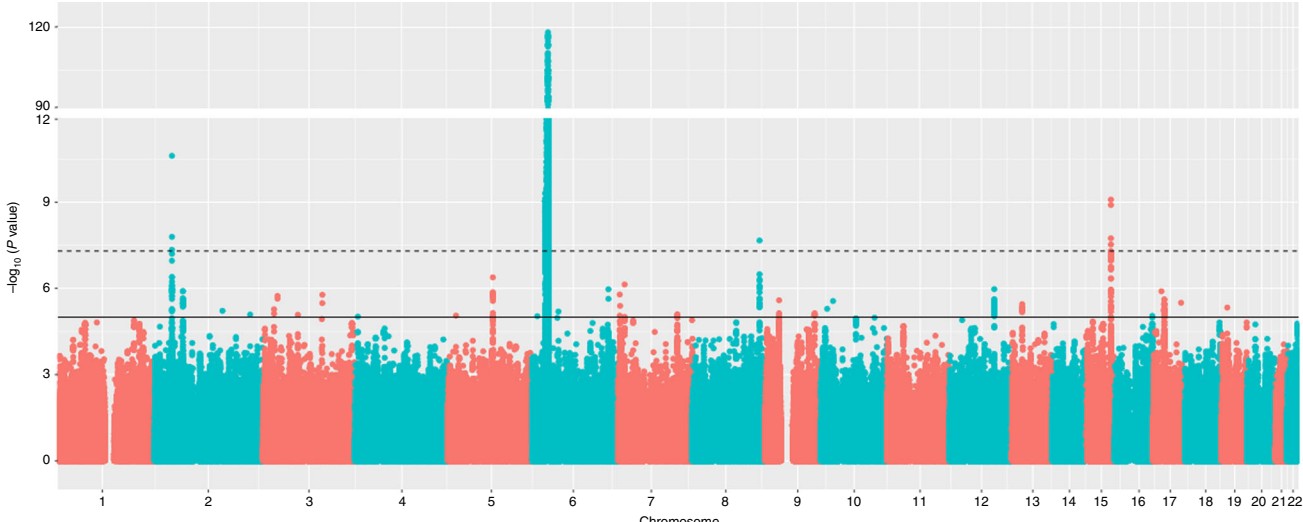

**Fig. 2** Manhattan plot showing the P values for the meta-analysis genome-wide association study. Each SNP was tested for association by logistic regression using an additive regression model; the interrupted line indicates the threshold for genome-wide significance ($P = 5 \times 10^{-8}$); the y axis has been collapsed for better illustration of all genomic signals; the continuous line represents the threshold for suggestive significance ($P = 1 \times 10^{-5}$); $N = 5161$ biologically independent subjects ($N_{cases} = 1044$ and $N_{controls} = 4145$)

the role of HLA genes in FFA is complicated by the complex linkage disequilibrium structure across the region, a sequential conditional analysis with both SNPs and imputed HLA alleles indicates that there may be at least a further two independent HLA-alleles that contribute independently to disease risk (Supplementary Figure 3 and Supplementary Tables 4A and 4B).

To further investigate causal genes and alleles at the three remaining FFA susceptibility loci, we performed Bayesian fine-mapping of the association signals. This process identified a single putative causal variant with a posterior probability >0.5 of being the causal variant underlying the association signal at two of the three loci (Fig. 3 and Supplementary Table 5). At 2p21.2, rs1800440, is likely to be the causal variant underlying the association signal with a posterior probability of 0.98 (Fig. 4). The FFA protective allele is a missense allele (c.1358A>G p. Asn453Ser) in the CYP1B1 gene, which introduces a serine residue in the haem binding domain of the enzyme. In silico pathogenicity prediction tools predict that this allele has a deleterious effect on the function of the protein (SIFT = 0.015; CADD = 32)[13,14], which is corroborated by published functional investigations of the p.Asn453Ser substitution[15]. At the 8q24.22 locus, rs760327 is the most likely causal allele (posterior probability = 0.68). The variant is located within intron 1 of the ST3GAL1 gene encoding the homonymous galactoside sialyl-transferase enzyme, which has been studied in the context of T cell homeostasis[16,17]. At 15q.26.1, statistical fine mapping was unable to clearly resolve the causal variant at this locus (Supplementary Table 5), though the most likely causal variant rs34560261 (posterior probability = 0.4) is located within intron 1

of SEMA4B. Co-localization with skin eQTLs (Supplementary Figure 5) provides evidence that the same variant(s) underlying the observed FFA association at this locus are also associated with variation in the expression of SEMA4B in the skin ($P_{coloc} = 0.99$), providing support that SEMA4B may be the causal gene at this locus.

**Plasma metabolomic analysis.** At the 2p22.2 locus statistical fine mapping of the causal allele to a functional missense variant in CYP1B1 implicates variation in xenobiotic and endogenous hormone metabolism[18–22] as a potential mechanism influencing disease susceptibility. To evaluate if there are systematic differences in metabolomic profiles between FFA cases and controls we compared levels of plasma metabolites in 52 treatment-naïve FFA cases and 35 ethnicity-, gender-, age- and BMI-matched healthy controls (Supplementary Figure 6). We did not observe differences in the distribution of individual metabolite levels between cases and controls of the magnitude detectable by this experiment following multiple-testing correction (Supplementary Table 6), nor did we observe any enrichment of xenobiotic or endogenous hormone metabolites in the extremes of the distribution of observed mean differences.

**Transcriptomic analysis.** To further investigate genes and biological pathways involved in FFA pathobiology we performed transcriptome sequencing in lesional scalp skin from seven treatment-naïve postmenopausal FFA cases and compared to transcriptome profiles from scalp skin in seven

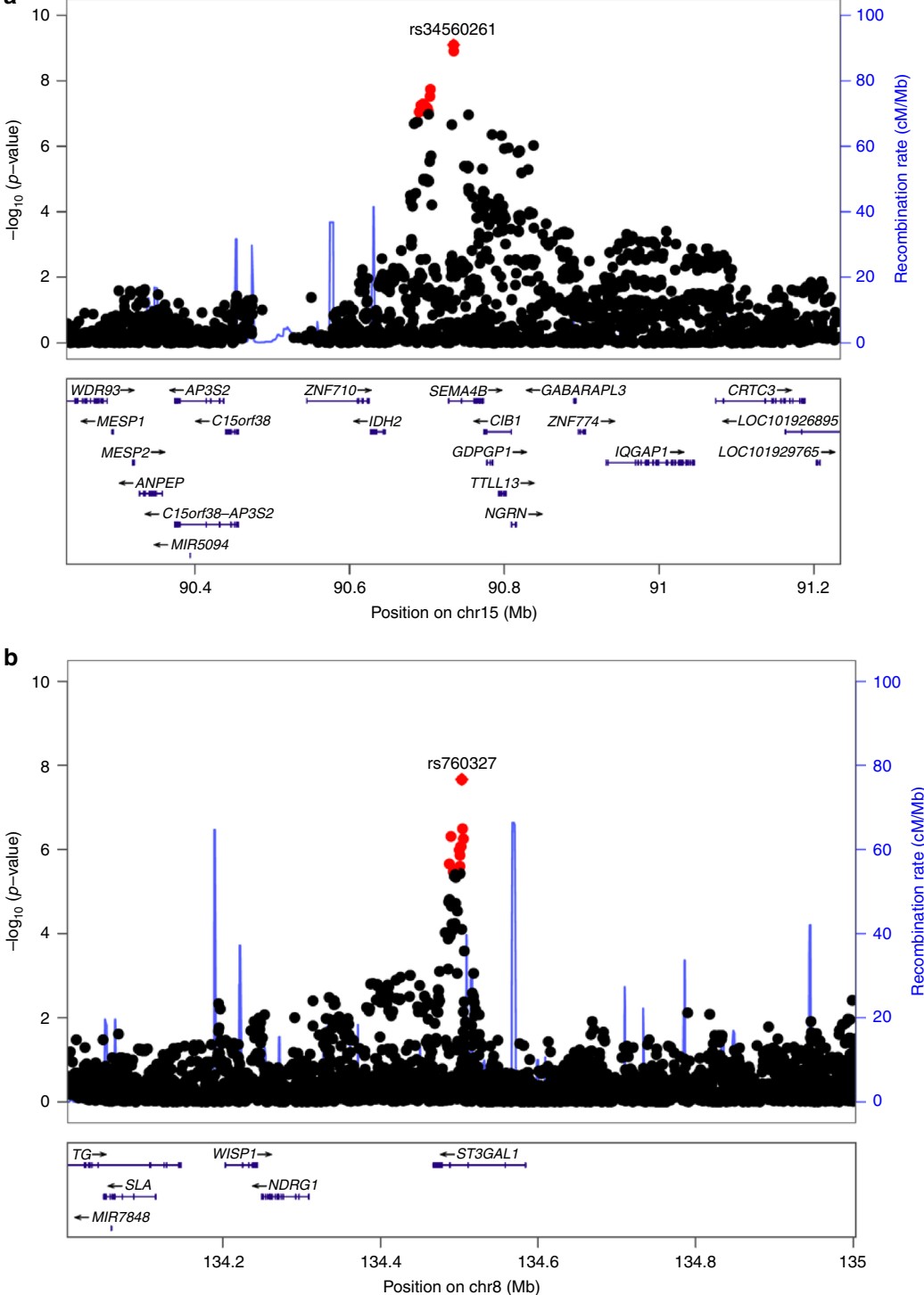

**Fig. 3** Regional plots of lead signals at loci 15q26.1 and 8q24.22. **a** Regional plot at locus 15q26.1 (rs34560261); and **b** Regional plot at locus 8q24.22 (rs760327). The blue lines shows the fine-scale recombination rates (right y axis) estimated from individuals in the 1000 Genomes population; genes are highlighted with horizontal lines; the x axis shows the chromosomal position in Mb; variants within the 95% credible set are highlighted in red

healthy matched controls. Differences in transcript abundance between cases and controls in these bulk tissue samples was observed for 117 genes with a greater representation of transcripts from 80 genes in affected tissue and 37 genes with reduced representation of their respective transcripts (Supplementary Tables 7–9; Supplementary Figure 7). Of these, only *C2*, within the MHC at 6p21.33, is located within 1 Mb of any of the FFA associated loci and only two of the 117 genes

(*CCL19* and *EPSTI1*) are located within 1 Mb of any variant with moderate evidence of association ($P < 5 \times 10^{-5}$) with FFA. Investigation of the enrichment of gene sets and pathways indicate that immune genes are over-represented amongst the differentially expressed genes (DEGs) (Supplementary Tables 8–12). Notably, four of the 10 most extreme DEGs are genes that have an established role in the interferon gamma (IFNγ) pathway.

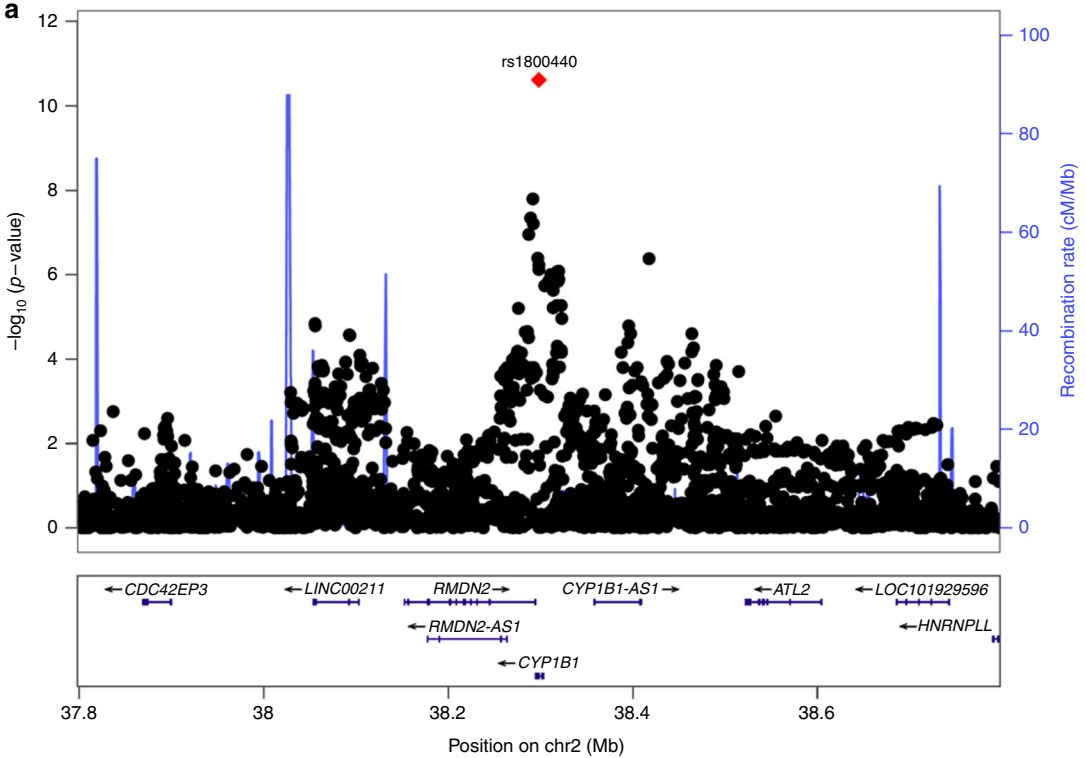

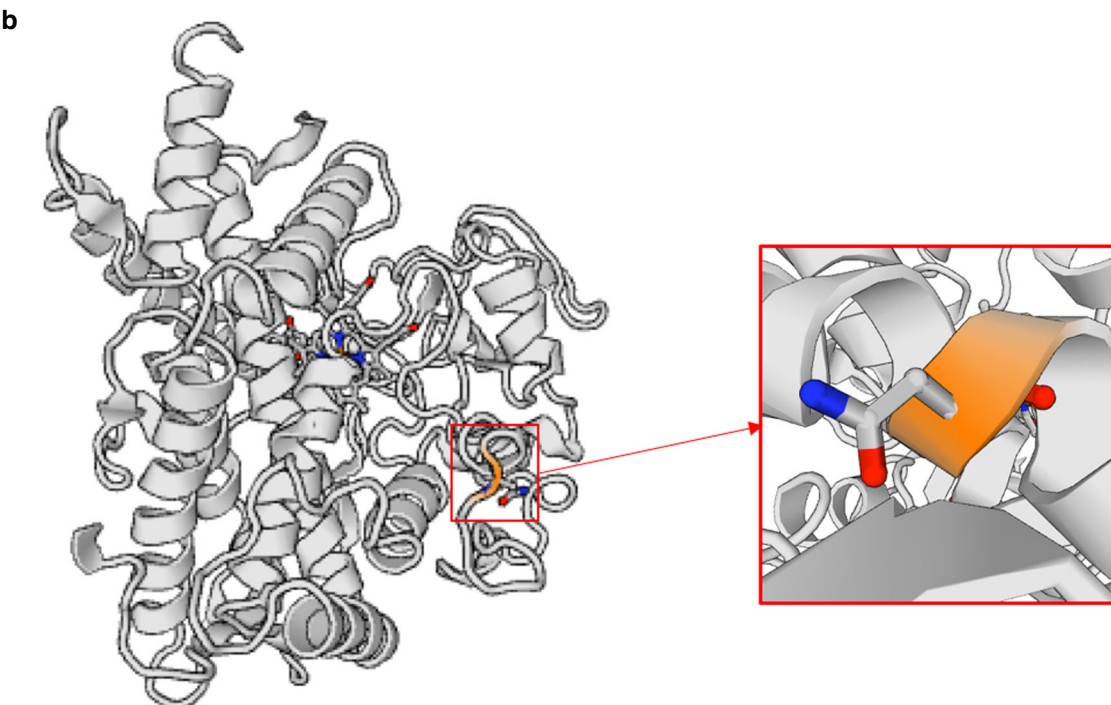

**Fig. 4** Regional plot at locus 2p22.2 and CYP1B1 enzyme structure. **a** Regional plot of lead signal at locus 2p22.2 demonstrating the lead causal variant (rs1800440); the blue lines shows the fine-scale recombination rates (right y axis) estimated from individuals in the 1000 Genomes population; genes are highlighted with horizontal lines; the x axis shows the chromosomal position in Mb; variants within the 95% credible set are highlighted in red ($N = 1$, posterior probability = 0.98); **b** CYP1B1 enzyme structure drawn via the SWISS-MODEL repository with the site affected by the residue change from Asparagine (N) to Serine (S) at position 453 magnified (https://swissmodel.expasy.org/repository)

## Discussion

We have identified four genomic loci, at which genetic variation is robustly associated with the lichenoid inflammatory and scarring dermatosis FFA in two independent cohorts of European ancestry.

The strongest effect on FFA susceptibility is observed at 6p21.1 which is located within the *MHC* region. Through imputation of classical *HLA* alleles we implicate the Class I allele *HLA-B*07:02* as conferring a five-fold increase in risk of FFA. The highly polymorphic *HLA* genes and their encoded proteins play a key

role in self and non-self immune recognition and are known to determine susceptibility to numerous infectious and auto-immune disorders[23]. *HLA-B*07:02* itself has previously been reported to be associated with HIV progression but has not been implicated in the susceptibility to human disease[24]. The hair follicle bulge region and the outer root sheath express low levels of *HLA-A*, *HLA-B* and *HLA-C* and these are key to rendering immune privilege[25,26]. *HLA-B*07:02* may facilitate the process of hair follicular autoantigen presentation culminating in the auto-inflammatory lymphocytic destruction of the hair follicle bulge and its resident epithelial hair follicle stem cells. Investigation of differentially expressed genes between affected and unaffected scalp tissue further highlights the importance of genes encoding the components of innate and adaptive immunity and, notably, the IFNγ pathway, which is an important regulator of antigen presentation. Also relevant to a putative role of T cell dysfunction in FFA, the lead variant at the 8q24.22 locus is located in intron 1 of *ST3GAL1*, which encodes a membrane bound sialyltransferase. Changes in cell surface glycan structures have been implicated in human T cell lymphocyte activation and maturation[27] and *ST3GAL1* itself has been implicated in immunity by home-ostatically controlling CD8$^+$T cells[16,17].

At 2p22.2 we observe strong evidence that the causal variant underlying the association at this locus is a missense variant in *CYP1B1*, a ubiquitously expressed gene encoding the Cytochrome P450 1B1 microsomal enzyme, also known as xenobiotic mono-oxygenase and aryl hydrocarbon hydroxylase. This enzyme contributes to the oxidative metabolism of oestradiol and oestrone to their corresponding hydroxylated catechol oestrogens[28–30]. Functional investigation of allelic variation in *CYP1B1* has shown that the FFA protective p.Asn453Ser allele increases the rate of CYP1B1 degradation leading to reduced intracellular CYP1B1 levels[15]. Given the established role of CYP1B1 in sex hormone metabolism, alongside the female preponderance of FFA and its rapid and recent increase in incidence, it is plausible to speculate that an increase in exposure of females to a CYP1B1 substrate, whether endogenous or exogenous, may contribute to the development of FFA. The temporal relationship between the introduction of oral contraceptives in the 1960s and the appearance of FFA in the published literature in the 1990s should be fully explored with a well-designed gene-environment inter-action study. Nevertheless, no striking differences in such sub-strates nor any other metabolites were identified in our metabolomic study although this may reflect the limited power of this initial investigation to observe more subtle disruption of this metabolic pathway. It should be noted that *CYP1B1* has also been implicated in human immune cell regulation[31,32] and the potential that FFA susceptibility at the 2p22.2 is mediated through immune pathways cannot be excluded.

In summary, in this exploration of the molecular genetics of FFA susceptibility we identify common alleles at four genomic loci that contribute to disease risk. The putative biological impact of this genetic variation indicates that the disease is a complex immuno-inflammatory trait underpinned by risk alleles in MHC Class I molecule-mediated antigen processing and T cell home-ostasis and function. The insight into the pathobiology of FFA from the genetic susceptibility loci combined with the observation that there is an increase in transcripts encoding components of the IFNγ pathway in affected scalp tissue suggest that drugs such as JAK inhibitors, some of which have already proved effective in alopecia areata[33] and trialled in lichen planopilaris[34], may prove to be efficacious for FFA.

## Methods

**Clinical resource**. Ethical approval was granted by the Northampton NRES Committee, UK (REC 15/EM/0273) and the study was conducted in accordance with the Declaration of Helsinki (https://www.wma.net/policies-post/wma-declaration-of-helsinki-ethical-principles-for-medical-research-involving-human-subjects/). We ascertained two independent cohorts of highly clinically consistent female cases of classic FFA, diagnosed by specialist dermatology clinics in the UK and Spain. All recruited cases were of European ancestry and diagnosed with FFA on the basis of the following clinical and histopathological features (recently proposed as diagnostic criteria)[35]: (1) cicatricial alopecic involvement of the frontal, temporal/parietal hair margin; (2) bilateral eyebrow loss; (3) clinical, tri-choscopic (or histological) evidence of lichenoid perifollicular inflammatory pre-sence; (4) facial or body hair loss; (5) absence of multifocal scalp involvement and other signs suggestive of classic LPP or its Graham-Little-Piccardi-Lasseur subvariant.

All research participants provided written informed consent for participation in the study. The individual depicted in Fig. 1 provided informed consent for publication of her clinical images.

**Genotyping and genome-wide association analysis**. For the UK cohort, genome-wide genotyping of cases was undertaken using Infinium OmniEx-pressExome BeadChip array (Illumina) and an unselected female control cohort from the English Longitudinal Study of Aging (ELSA) project (http://www.elsa-project.ac.uk), genotyped on the Infinium Omni2.5M BeadChip array (Illumina). We retained variants that were assayed with the same probe design on both gen-otyping arrays and excluded variants with a call rate of <99% or which deviated from Hardy–Weinberg Equilibrium ($P < 10^{-4}$). Individuals with a call rate of <99% or extensive heterozygosity were also excluded. A subset of 46,789 variants in linkage equilibrium ($r^2 < 0.2$ between each pair) was used to evaluate relatedness between individuals using the KING software package (KING; version 2.1.1). Individuals were thus removed from the study such that no two individuals had estimated relatedness closer than third degree (Kinship coefficient > 0.0442). Principal component analysis of the same set of 46,789 variants was performed and individuals outlying the main cluster (implying non-European ancestry) were also excluded from further analysis.

In the Spanish cohort, FFA case genotyping was performed using the Infinium OmniExpressExome BeadChip array (Illumina). Genotypic data for unaffected controls were obtained from 1061 individuals from the INfancia y Medio Ambiente (INMA) project (Valencia, Sabadell and Menorca, Spain http://www.proyectoinma.org) genotyped on the Omni1-Quad BeadChip (Illumina). Genotype calling, quality control and imputation were undertaken following the same protocol as described for the UK cohort across all variants that are assayed on both genotyping arrays with the same probe design.

Following quality control, genome-wide imputation was performed for both cohorts using the Michigan Imputation Server, with the Haplotype Reference Consortium (HRC) reference haplotype panel (www.haplotype-reference-consortium.org). All variants with an imputation info score <0.7 or a minor allele frequency of <0.005 were excluded from downstream analysis. This process of data generation and QC resulted in a combined total of 7,039,930 variants successfully genotyped or imputed in a combined total of 1016 cases and 4145 controls.

Genome-wide association testing was performed on all variants with MAF > 0.005 using a logistic Wald association test (EPACTS), including the first five principal components as covariates. Association testing was performed based on 844 affected females and 3760 female controls from the UK cohort and separately for 172 affected females and 385 female controls from the Spanish cohort. Association summary statistics were subsequently combined for 7,039,930 variants across the UK and Spanish cohorts via a standard error-weighted meta-analysis using METAL[36]. To evaluate potential confounding bias due to population stratification or residual cryptic relatedness we calculated the genomic inflation factor ($\lambda_{GC}$) for variants with MAF > 0.05 and the LD score regression intercept for each cohort and the combined meta-analysis[37].

**Causal variant identification and evaluation**. For the UK and Spanish case-control cohorts, imputation of classical HLA alleles to two- and four-digit reso-lution was performed with the SNP2HLA tool, based on the genotypes of 1297 MHC region single nucleotide polymorphisms (SNPs) genotyped in both phases[38]. In the UK cohort, dosage-based association testing was performed in PLINK v1.9 for all 208 alleles that were well imputed ($r^2 > 0.9$), using a logistic regression framework that included the same covariates as the full GWAS[39]. Replication of specific variants of interest was undertaken in the Spanish replication cohort in the same way, and standard-error weighted meta-analysis was performed using the meta package in R[40]. To test for multiple independent association signals, stepwise conditional analysis was performed: at each round of testing, the dosage of the HLA allele achieving the lowest discovery phase association p-value in the previous round was added to the list of covariates. This process was iterated until no allele achieved genome-wide significance ($P_{Meta} < 5 \times 10^{-8}$). To ascertain which specific variants underlie the observed allelic associations, a similar stepwise analysis was performed using imputed HLA-region SNPs in place of HLA alleles. To verify that no additional genome-wide significant independent signals remained after the final iteration, we also tested for association of the HLA-region SNPs in the full GWAS dataset after conditioning on all independently-associated HLA alleles.

At 2p22.2, 8q24.22 and 15q26.1, fine-mapping was undertaken to identify putative causal variant(s) underlying the observed association signal[41]. For each

locus, we constructed a credible set of variants considered most likely to be causal based on evidence for association as quantified by their Bayes factor[42].

In order to explore the correlation between genetic variation and tissue expression, eQTL colocalization analysis was performed between the observed FFA association signals and sun-exposed skin cis-eQTL data from the Gene-Tissue Expression (GTEx) project database[43]. Candidate skin eQTLs were identified by looking into whether any variant within the FFA risk loci was associated with varied expression of nearby genes. Bayesian testing for colocalization between the FFA association signal and the skin eQTL signal was undertaken using a set of overlapping variants for the two datasets, employing the R package coloc tool[44], with a defined prior probability of colocalization of $P = 10^{-5}$.

**Heritability estimation**. FFA heritability explained by genome-wide SNPs (MAF > 0.01) was estimated using the genomic-relatedness-based restricted maximum-likelihood (GREML) approach, implemented in the software tool package GCTA[45]. Heritability estimates were expressed on the liability scale using an estimated prevalence of FFA of one in 5000.

**Transcriptomic analysis**. We performed transcriptome profiling with RNA-sequencing of scalp skin from seven cases of European ancestry and seven matched controls (Supplementary Table 1). All seven cases were clinically evaluated prior to obtaining skin biopsy from actively inflamed parietal scalp skin for histologic confirmation of FFA. Samples from cases were only subjected to downstream processing if they were confirmed to be actively inflamed upon histological evaluation. Macroscopically unremarkable parietal scalp skin was also harvested from healthy controls undergoing plastic facial surgery and all control tissue specimens were also examined microscopically and confirmed to be histologically normal (Supplementary Figure 1). Total RNA was isolated from each tissue sample using the RNeasy Plus Universal kit (Qiagen, Valencia, USA) as per the manufacturer's protocol and instructions. Samples with RNA Integrity Number (RIN) < 8 were rejected from further processing.

Whole transcriptome RNAseq libraries were prepared using the Agilent SureSelect strand-specific RNA library preparation kit (Agilent, Santa Clara, USA) and multiplexed sequencing was performed on the HiSeq 2500 platform (Illumina, San Diego, USA).

Processing of the raw transcriptomic data files was conducted using an established analytical pipeline (Supplementary Figure 2). EdgeR software package in the (R-based) Bioconductor was utilized to undertake differential expression analysis[46], following the trimmed mean of M-values (TMM) normalization method[47]. Genes with very low expression (defined as genes with counts per million (CPM) < 1 in at least seven samples) were discarded and not considered for further differential expression analysis. Transcript abundance was estimated and compared between the two groups with an exact (Robinson & Smyth) negative binomial (NB) test in EdgeR (Supplementary Figure 2). The P value distribution was obtained and Benjamini-Hochberg (BH) adjusted P values were estimated: genes with a false discovery rate FDR ≤ 0.05 and log fold change LFC ≥ 1 or ≤ −1 were considered significantly differentially expressed and extracted. Such differentially expressed genes were used to generate a heat map using the R package gplots2 via the START interface[48].

Gene set enrichment and pathway analysis (GSEA) was performed as implemented in GAGE (R package) using C2 (pathways) and C5 (gene ontology) gene set collections from the Molecular Signatures Database (MSigDB)[49,50]. The p values for the gene set enrichment were calculated using default Stouffer test in GAGE and FDRs were generated using the BH procedure. Differentially expressed HLA genes were excluded from the analysis because of the challenges of accurately quantifying the expression levels of these highly polymorphic genes (due to the difficulty of correctly mapping divergent reads to a single reference genome).

**Plasma metabolomic analysis**. Fifty-two treatment-naïve post-menopausal female FFA cases of European ancestry (median age 64; mean BMI 24.7) and 35 matched controls (median age 58; mean BMI 24.5) were recruited. Peripheral venous blood was collected and centrifuged (at 1300 g for 15 min) to separate plasma, which was aliquoted and stored at −80 ºC until required for further analysis. Metabolomic profiling of samples was undertaken by Metabolon (Durham, NC, USA) by subjecting plasma samples to methanol extraction prior to splitting into aliquots for analysis by ultra-high pressure liquid chromatography/mass spectrometry (UHPLC/MS)[51]. Metabolites were identified by automated comparison of ion features to a reference library of chemical standards followed by visual inspection for quality control[52]. Missing values were presumed to be below the limits of detection and were therefore imputed to the compound minimum.

Metabolomic data analysis was undertaken, having accounted for medicinal drug-related by-products and discarded unnamed molecules. We constructed a heat map illustrating group differences at individual and group average level using MetaboAnalyst 4.0[53]. Univariate comparison of abundance of 947 named small-molecule (<500Da) metabolites between cases and controls was performed using the Mann-Whitney U test.

## Data availability

Full meta-analysis summary statistics are available at the European Genome-phenome Archive (EGA) under the collection ID EGAS00001003460. All raw and processed transcriptomic data are available at the Gene Expression Omnibus (GEO) under the collection ID GSE125733. All other data that support the findings of this study are available from the corresponding author upon reasonable request.

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

## Acknowledgements

We acknowledge financial support from the Department of Health via the National Institute for Health Research (NIHR) Rare Diseases Translational Research Collaboration (NIHR-RD TRC) and by the NIHR Biomedical Research Centre based at Guy's and St Thomas' NHS Foundation Trust and King's College London. Samples and data from Spanish patients included in this study were provided by the Hospital Universitario Ramón Y Cajal-IRYCIS Biobank integrated in the Spanish Hospital Platform Biobanks Network and were processed following standard operation procedures with appropriate approval of the Ethical and Scientific Committees. We thank the study participants in the UK and Spain for their help. Data used for replication in this research was provided by the INMA—INfancia y Medio Ambiente [Environment and Childhood] Project (www.proyectoinma.org), which is supported in part by funds. We are thankful to the English Longitudinal Study of Ageing (ELSA) for providing population control genotyping data. We thank the NIHR Rare Genetic Disease Research Consortium Agreement Team, especially Gillian Borthwick, for their help with setting up multiple UK research participating sites. We thank Dr. Sophia Karagiannis, Dr. Francesca Capon, Prof Jemma Mellerio and Dr. Ian White for their advice and guidance. We are thankful to Prof Kathryn Lewis, Dr. Raquel Iniesta, Dr. Ken Hanscombe and Dr. Leo Bottolo for their input and advice with the statistical analysis aspects of the metabolomic exploration. We thank the numerous research assistants and nurses, especially Teena Mackenzie, Sophie Devine, Ruth Joslyn, Sonia Baryschpolec, Anne Thomson, Pauline Buchanan and Caroline White for their help with recruitment. Very special thanks go to Dr. Dimitra Dritsa for her continuous support and input throughout this work.

## Author contributions

C.T., M.A.S and J.A.M. designed the study and wrote the manuscript, C.T. orchestrated multi-site patient recruitment and obtained institutional ethics approval. C.T., C.P., M.A.S., J.A.M., N.D., C.A., J.R.S., V.P., Ed.R, G.M. and S.S. performed the analysis and interpretation of data. C.T., R.P., D.B., A.O., C.J.C., L.L., S.M.L., E.A.A.C., K.W. and A.S. contributed to laboratory work. D.A.F., S.H., F.C., G.P., M.H., I.P., M.K., P.F., A.Mc, A.M., J.J., V.J., I.A., M.A.-J., C.M., H.C., N.B., R.A., C.B., A.A., C.C., S.V-G., A.M.M.R.; N.O.P., G.K.P., M.P., A.Mc, A.B., M.M., S.W., K.A., N.C., M.G., I.M., D.d.B., G.D., A.T., A.R., T-W.S., R.S., M.S.W., N.D., N.K., J.S., F.L., C.M.S. and K.B. contributed to patient phenotyping and recruitment. X.E. provided access to population control genotyping data. D.A.F. oversaw research clinic activities and patient phenotyping. All authors contributed to revising the manuscript and made contributions to the study design, data acquisition, data analysis and interpretation.

## Additional information

**Competing interests:** The authors declare no competing interests.

Christos Tziotzios [1], Christos Petridis[2], Nick Dand [2], Chrysanthi Ainali[1], Jake R. Saklatvala[2], Venu Pullabhatla[3], Alexandros Onoufriadis[1], Rashida Pramanik[3], David Baudry[1], Sang Hyuck Lee[4], Kristie Wood[3], Lu Liu[5],

Seth Seegobin[1], Gregory A. Michelotti[6], Su M. Lwin[1], Evangelos A.A. Christou[1], Charles J. Curtis[3], Emanuele de Rinaldis[3], Alka Saxena[3], Susan Holmes[7], Matthew Harries[8], Ioulios Palamaras[9], Fiona Cunningham[7], Gregory Parkins[7], Manjit Kaur[10], Paul Farrant[11], Andrew McDonagh[12], Andrew Messenger[12], Jennifer Jones[13], Victoria Jolliffe[14], Iaisha Ali[15], Michael Ardern-Jones [16], Charles Mitchell[17], Nigel Burrows[18], Ravinder Atkar[18], Cedric Banfield[19], Anton Alexandroff[20], Caroline Champagne[21], Hywel L. Cooper[17], Sergio Vañó-Galván[22], Ana Maria Molina-Ruiz[23], Nerea Ormaechea Perez[24], Girish K. Patel[25], Abby Macbeth[26], Melanie Page[26], Alyson Bryden[27], Megan Mowbray[28], Shyamal Wahie[29], Keith Armstrong[30], Nicola Cooke[31], Mark Goodfield[32], Irene Man[33], David de Berker[34], Giles Dunnill[34], Anita Takwale[35], Archana Rao[36], Tee-Wei Siah[37], Rodney Sinclair[38], Martin S. Wade[39], Ncoza C. Dlova[40], Jane Setterfield[1], Fiona Lewis[1], Kapil Bhargava[1], Niall Kirkpatrick[41], Xavier Estivill [42], Catherine M. Stefanato[43], Carsten Flohr[1], Timothy Spector[44], Fiona M. Watt [45], Catherine H. Smith[1], Jonathan N. Barker[1], David A. Fenton[1], Michael A. Simpson[2] & John A. McGrath [1]

[1]St. John's Institute of Dermatology, King's College London, London, Guy's Hospital, London SE1 9RT, UK. [2]Department of Medical and Molecular Genetics, King's College London, Guy's Hospital, London SE1 9RT, UK. [3]UK NIHR GSTFT/KCL Comprehensive Biomedical Research Centre, Guy's & St. Thomas' NHS Foundation Trust, London SE1 9RT, UK. [4]NIHR Maudsley Biomedical Research Centre at South London and Maudsley NHS Foundation Trust (SLaM) & Institute of Psychiatry, Psychology and Neuroscience (IoPPN), King's College London, Denmark Hill Campus, London SE5 8EF, UK. [5]National Diagnostic EB Laboratory, St. Thomas Hospital, London SE1 7EH, UK. [6]Metabolon, Inc., Morrisville, NC 27560, USA. [7]Alan Lyell Centre for Dermatology, Queen Elizabeth University Hospital, Glasgow G51 4TF, UK. [8]The Dermatology Centre, The University of Manchester, Salford Royal NHS Foundation Trust, Salford M6 8HD, UK. [9]Department of Dermatology, Barnet General Hospital, Royal Free Foundation Trust, London EN5 3DJ, UK. [10]Department of Dermatology, Solihull Hospital, Solihull B91 2JL, UK. [11]Department of Dermatology, Brighton and Sussex University Hospitals NHS Trust, Brighton BN2 3EW, UK. [12]Department of Dermatology, Royal Hallamshire Hospital, Sheffield S10 2JF, UK. [13]Department of Dermatology, Royal Free Hospital, London NW3 2QG, UK. [14]Department of Dermatology, Royal London Hospital, Barts Health NHS Trust, London E1 1BB, UK. [15]Department of Dermatology, Imperial College Healthcare NHS Trust, London W12 0HS, UK. [16]Department of Dermatology, University Hospitals Southampton NHS Foundation Trust, Southampton SO16 6YD, UK. [17]Department of Dermatology, Portsmouth Hospitals NHS Trust, Portsmouth PO6 3LY, UK. [18]Department of Dermatology, Addenbrooke's Hospital, Cambridge University Hospitals NHS Foundation Trust, Cambridge CB2 0QQ, UK. [19]Department of Dermatology, Peterborough City Hospital, Peterborough PE3 9GZ, UK. [20]Department of Dermatology, University Hospitals of Leicester, Leicester Royal Infirmary, Leicester LE3 9QP, UK. [21]Department of Dermatology, The Churchill Hospital, Oxford OX3 7LE, UK. [22]Trichology Unit, Dermatology Department, Ramon Y Cajal Hospital, University of Alcala, IRYCIS, Madrid 28034, Spain. [23]Hospital Universitario Fundación Jiménez Díaz, Madrid 28040, Spain. [24]Department of Dermatology, Hospital Donostia, San Sebastian 20014, Spain. [25]Hywel Dda University Health Board, Cardiff SA31 3BB, UK. [26]Department of Dermatology, Norfolk & Norwich University Hospitals NHS Foundation Trust, Norwich NR4 7UY, UK. [27]Department of Dermatology, Ninewells Hospital, Dundee DD1 9SY, UK. [28]Department of Dermatology, Queen Margaret Hospital, Dunfermline KY12 0SU, UK. [29]Department of Dermatology, County Durham and Darlington NHS Foundation Trust, Darlington DL3 6HX, UK. [30]Department of Dermatology, Royal Victoria Hospital, Belfast BT12 6BA, UK. [31]Department of Dermatology, Whiteabbey Hospital, Northern Health & Social Care Trust, Co Antrim BT37 9RH, UK. [32]Department of Dermatology, Chapel Allerton Hospital, Leeds teaching Hospitals NHS Trust, Leeds LS7 4SA, UK. [33]Department of Dermatology, Surrey and Sussex Healthcare NHS Trust, Surrey RH1 5RH, UK. [34]Department of Dermatology, University Hospitals Bristol NHS Foundation Trust, Bristol BS2 8HW, UK. [35]Department of Dermatology, Gloucestershire Hospitals NHS Foundation Trust, Gloucester GL1 3NN, UK. [36]Department of Dermatology, Kingston Hospital NHS Foundation Trust, Kingston-upon-Thames KT2 7QB, UK. [37]Department of Dermatology, Royal Victoria Infirmary, Newcastle upon Tyne NE1 4LP, UK. [38]Department of Dermatology, University of Melbourne, Melbourne VIC 3010, Australia. [39]The London Skin and Hair Clinic, London WC1V 7DN, UK. [40]Department of Dermatology, School of Clinical Medicine, Nelson R Mandela School of Medicine, Durban 4001, South Africa. [41]Craniofacial Surgery Unit, Chelsea & Westminster Hospital, London SW10 9NH, UK. [42]Genetics Unit, Dexeus Women's Health, Barcelona 08028, Spain. [43]Department of Dermatopathology, St. John's Institute of Dermatology, St. Thomas' Hospital, London SE1 7EH, UK. [44]The Department of Twin Research & Genetic Epidemiology, King's College London, London SE1 7EH, UK. [45]Centre for Stem Cells and Regenerative Medicine, King's College London, Guy's Hospital, London SE1 9RT, UK. These authors contributed equally: Michael A. Simpson, John A. McGrath.

