## [Peer Review File · Nature Communications]

Reviewer #1 (Remarks to the Author):

This is a well-conducted and well-written report of a GWAS and gene expression experiment providing robust biological insight into FFA. My main concern with the paper is the rationale for and interpretation of the metabolomics data.

CYPB1 is a well-characterized tumor antigen and CYP1B1 peptides have been shown to be presented by HLA-A and serve as targets for antigen-specific cytotoxic T cells. Thus, rather than exerting an effect through metabolic activity, CYPB1 could be influencing an immune response targeted to the HF, a hypothesis that is much better supported by the other GWAS loci and their gene expression evidence. In fact, the data from their metabolomic profiles also better supports an immunological role of CYPB1, given that no significant differences were detected in metabolite levels and there is not a clear link between the metabolites with suggestive evidence and CYPB1. Given their findings, it is misleading to have this claim in the title. The genetic and genomic data clearly supports FFA as an immune-inflammatory disorder, the link to xenometabolism is tenuous.

Their rationale and interpretation of the metabolomics data needs to be toned down. For example “Plasma metabolomic profiles in FFA further indicate that metabolomic aberrations are of relevance in FFA disease pathogenesis and this warrants further exploration” is overstating the suggestive evidence generated from their data. They did not observe any statistically significant difference in metabolites between cases and controls. Do the metabolites that they identified with their random forest classifier serve as substrates for CYP1B1?

The metabolomics data is not convincing and difficult to interpret. For example, it is difficult to see striking difference between cases and controls in Figure S7, although labels indicating cases and controls could help and a way to map the compounds to Table S17 would help.

The statement “CYP1B1 (see Results) has an established role in xenobiotic and hormonal metabolism and the FFA protective p.Asn453Ser allele has been demonstrated experimentally to disrupt the enzymatic activity of the encoded cytochrome P450 1B1” should have a reference. In the following sentence, the word highlights should be toned down to suggests: “This highlights variation in xenobiotic and hormone metabolism as a potential mechanism influencing disease susceptibility.”

HLA-B*07:02 has been associated with HIV progression. The statement that “This particular HLA-B allele has not, to our knowledge, previously been reported to be associated with any other disease.” should be removed.

Additional concerns and suggestions:

Table 1 should contain all independent significant associations including the SNPs that underlie the associations of classical alleles HLA-A*33:01 and HLA-A*11:01. It is possible that there are multiple disease effects arising from the HLA, and in fact their conditional analysis of classical alleles demonstrates this. Additionally, because of the complex linkage disequilibrium structure across the HLA, they should report all of the significant SNPs in a supplementary table. This would facilitate independent investigation of effects conferred by long-range haplotypes.

It is unclear if the supplemental table 2 includes all SNPs for $p < 1 \times 10^{-5}$, or just one per locus. They should list all. This would permit a better understanding of the size of the locus and a precise definition of the associated haplotype, which would be useful for integrating into future sequencing studies and would also facilitate annotation of disease associated SNPs for functional relevance.

The supplement for the gene expression data should list all differentially expressed genes (N=120), not just 60 of them. Again, this would facilitate replication of their pathway analysis by independent investigators.

Did they check to see if any of these 120 genes are located within or near associated loci (either significant or suggestive)? Long range chromatin looping experiments demonstrate that risk SNPs can influence genes outside of the associated locus, up to 1Mb away (<https://www.nature.com/articles/ncomms10069>).

I can't see that Tables S8 and S10 add anything.

Reviewer #2 (Remarks to the Author):

This is a thoroughly performed study presented in a very well written manuscript. The identification of HLA risk alleles, and three other interesting genetic associations provide novel information to contribute to understanding mechanisms of FFA.

The identification of these genes is made even more interesting by the metabolomics results which tie into the identification of the CYP1B1 protective missense variant.

I have some very minor suggestions and questions:

The overrepresentation of immune genes in cases in the RNA-seq results seems like a very obvious result, and perhaps this is why the authors have not overemphasized findings from the RNA-seq. Figure 1D shows lymphoid cell infiltrate of hair follicles in a patient, but there is no histopathology from a healthy control. Does one expect to see any lymphoid cells at all in a healthy control? If some lymphoid cells are present, the authors may consider using a tool like CYBER SORT to estimate the percentage of immune cells that contribute to the differentially expressed gene list, as a way to confirm the source of the overexpressed genes. Further to this, do the authors expect that the underexpressed genes in cases result from artifact due to overrepresentation of immune cell transcripts and subsequent underrepresentation of other cell's transcripts?

It seems that the results presented here implicate CD8 T cell pathology, as did GWAS into alopecia areata. Can the authors comment on the 'repurposing' of treatments effective in AA models and AA clinical trials for FFA?

The implication of the CYP1B1 missense mutation is fascinating. By which mechanism exactly do the authors think that 'reduced clearance of endogenous and exogenous substrates' in the presence of the CYP1B1 risk allele actually contribute to FFA development?

It is intriguing that metabolites pertinent to the xenobiotic super-pathway were identified in the metabolomics comparison of cases and controls. Approximately 10 of the controls should be heterozygous for the CYP1B1 missense mutation. Is there any evidence for xenobiotic metabolite differences between controls carrying the risk allele and those homozygous for the protective allele? Is there any evidence of xenobiotic metabolite differences in cases when controlling for the CYP1B1 genotype?

If the guidelines allow for more figures, it would be good to include Figures S5 and S8 as main figures, rather than supplementary. Table S3 should show the allele frequencies of the HLA-B*07:02 alleles in cases and controls.

Typos:

Line 33; "relevance xenobiotic processing" should read "relevance of xenobiotic processing"

Line 299; "been implicated immunity" should read "been implicated in immunity"

Line 307; "oestradiol to and oestrone to" should read "oestradiol and oestrone to"

Reviewer #3 (Remarks to the Author):

In this work, the authors presented a multi-omics study to study frontal fibrosing alopecia, and revealed 4 new disease loci. Here are my major questions:

* For either the UK or the Spanish cohort, the cases and controls were typed on different genotyping array. While only probes with same probe design were retained. I am a bit skeptical that potential difference in quality control/genotype calling performed separately in case/control will give rise to false positive results.

* The authors have to provide more details regarding the plausibility of applying the fine-mapping technique to the relatively low-sample size cohorts

* The authors should report the lambda GC values in result section. With the given sample size, and using markers with $MAF > 0.005$, it will not be surprising that the lambda GC looks "ok". I would suggest to compute lambda GC using common variants (e.g. $MAF > 0.05$).

* In Table 1, the OR from the meta-analysis for HLA-B looks very suspicious, as it is higher than both UK and Spanish cohorts, and it has a surprisingly sig. meta-p-value (when comparing with those from the UK and Spanish cohorts)

* Have the authors look into the overlap between the DEGs from RNA-seq to those loci with at least suggestive evidence of association?

* Are the patients from the metabolomic analysis the same as those from the RNA-seq ? In addition, is there any correlation between metabolite values with the gene expression profiles?

* While I appreciate the authors study the FAA using multi-omics approach, I find the results lack the coherence in presenting how the genetics, transcriptomics, and metabolomics results

* How did the authors ensure the robustness and generalizability in the RF classification ?

Dear Editor

We would like to thank you and the reviewers for their encouraging comments and constructive suggestions. These have been taken on board and the manuscript has been amended accordingly and we believe the changes have greatly strengthened the manuscript. Below is our point-by-point response to the reviewer's comments and attached is a revised manuscript with the respective changes highlighted.

We look forward to hearing from you in due course.

With many thanks and best wishes,

For and on behalf of the authors,

Prof Michael A. Simpson

Prof John A. McGrath

Reviewer #1 (Remarks to the Author):

This is a well-conducted and well-written report of a GWAS and gene expression experiment providing robust biological insight into FFA. My main concern with the paper is the rationale for and interpretation of the metabolomics data.

We thank the reviewer for their positive appraisal of our manuscript. The reviewer's concern relating the metabolomics experiment is well founded. Whilst this experiment constitutes only a minor component of the overall study we have re-approached the interpretation of the metabolomics data in light of this reviewer's comments below.

CYPB1 is a well-characterized tumor antigen and CYP1B1 peptides have been shown to be presented by HLA-A and serve as targets for antigen-specific cytotoxic T cells. Thus, rather than exerting an effect through metabolic activity, CYPB1 could be influencing an immune response targeted to the HF, a hypothesis that is much better supported by the other GWAS loci and their gene expression evidence. In fact, the data from their metabolomic profiles also better supports an immunological role of CYPB1, given that no significant differences were detected in metabolite levels and there is not a clear link between the metabolites with suggestive evidence and CYPB1. Given their findings, it is misleading to have this claim in the title. The genetic and genomic data clearly supports FFA as an immune-inflammatory disorder, the link to xenometabolism is tenuous.

We understand the reviewer's reluctance for the tentative link to xenometabolism to prominently feature in the title of the manuscript and have amended this appropriately and have toned down further speculation of the involvement of this mechanism in the discussion.

Their rationale and interpretation of the metabolomics data needs to be toned down. For example "Plasma metabolomic profiles in FFA further indicate that metabolomic aberrations are of relevance in FFA disease pathogenesis and this warrants further exploration" is overstating the suggestive evidence generated from their data. They did not observe any statistically significant difference in metabolites between cases and controls. Do the metabolites that they identified with their random forest classifier serve as substrates for CYP1B1? The metabolomics data is not convincing and difficult to interpret. For example, it is difficult to see striking difference between cases and controls in Figure S7, although labels indicating cases and controls could help and a way to map the compounds to Table S17 would help.

We accept the reviewer's concern relating to the metabolomics experiment. We have simplified our approach to both the analysis and interpretation of this data by reporting only the more easily interpreted univariate analysis of each metabolite, which as the reviewer points out, does not reveal any significant differences in any metabolite levels (that were measured by the assay) between cases and controls. We retain the figure illustrating the hierarchical clustering (Supplementary Figure 6) and have added case control labels and have removed the random forest classifier analysis to simplify the message in the results and discussion to indicate that no significant differences were observed.

The statement "CYP1B1 (see Results) has an established role in xenobiotic and hormonal metabolism and the FFA protective p.Asn453Ser allele has been demonstrated experimentally to disrupt the enzymatic activity of the encoded cytochrome P450 1B1" should have a reference. In the following sentence, the word highlights should be toned down to suggest: "This highlights variation in xenobiotic and hormone metabolism as a potential mechanism influencing disease susceptibility."

We have added the relevant citations and adjusted the wording of this part of the manuscript as suggested.

HLA-B*07:02 has been associated with HIV progression. The statement that "This particular HLA-B allele has not, to our knowledge, previously been reported to be associated with any other disease." should be removed.

We thank the reviewer for highlighting the link between HLA-B*07:02 and HIV progression, which we had missed. We have now added the relevant reference and additional text to the discussion.

Additional concerns and suggestions:

Table 1 should contain all independent significant associations including the SNPs that underlie the associations of classical alleles HLA-A*33:01 and HLA-A*11:01. It is possible that there are multiple disease effects arising from the HLA, and in fact their conditional analysis of classical alleles demonstrates this. Additionally, because of the complex linkage disequilibrium structure across the HLA, they should report all of the significant SNPs in a supplementary table. This would facilitate independent investigation of effects conferred by long-range haplotypes.

We agree that it is important to report the independent SNPs identified at this locus. However, in line with many previous GWAS studies we feel that the most appropriate presentation of results is for Table 1 to describe the loci identified via the main unconditional analysis (i.e. reporting the lead SNP per locus). We make it clear in our results section that for each locus the lead SNP may not be truly causal and may simply tag the causal genetic variation. This motivates additional analyses; within the HLA locus these include association testing for imputed classical HLA alleles and the identification of independent signals. Therefore, we feel the stepwise conditional analysis using SNPs naturally complements the stepwise conditional analysis of HLA alleles. As such, we include the results of both in Supplementary Table 4. While the complex relationship between SNPs and full HLA alleles means there is not a 1:1 correspondence, the SNP analysis supports the HLA alleles that were identified. Finally, full summary statistics for the full study, including the HLA region, are in the process of being uploaded in a publicly accessible repository.

It is unclear if the supplemental table 2 includes all SNPs for $p < 1 \times 10^{-5}$, or just one per locus. They should list all. This would permit a better understanding of the size of the locus and a precise definition of the associated haplotype, which would be useful for integrating into future sequencing studies and would also facilitate annotation of disease associated SNPs for functional relevance.

Supplementary Table 3 (previously Table S2) includes a single variant per locus (the variant with the strongest evidence of association). Rather than adding extensive tables to the supplementary materials, we are in the process of uploading a table with full summary statistics for the GWAS meta-analysis to the EGA repository, which will facilitate the analysis that the reviewer describes.

The supplement for the gene expression data should list all differentially expressed genes (N=120), not just 60 of them. Again, this would facilitate replication of their pathway analysis by independent investigators.

The complete list has now been included in the Supplementary Information (Supplementary Tables 8 and 9). In addition, the raw data for this experiment is in the process of being uploaded to EGA, so that independent investigators can have access.

Did they check to see if any of these 120 genes are located within or near associated loci (either significant or suggestive)? Long range chromatin looping experiments demonstrate that risk SNPs can influence genes outside of the associated locus, up to 1Mb away (<https://www.nature.com/articles/ncomms10069>).

We thank the reviewer for this suggestion. We have now looked to see if any of the top 120 differentially expressed genes were within 1 Mb of significant and suggestive associated regions and the results are incorporated in the results section of the manuscript.

I can't see that Tables S8 and S10 add anything.

We have removed these tables, as suggested.

Reviewer #2 (Remarks to the Author):

This is a thoroughly performed study presented in a very well written manuscript. The identification of HLA risk alleles, and three other interesting genetic associations provide novel information to contribute to understanding mechanisms of FFA. The identification of these genes is made even more interesting by the metabolomics results which tie into the identification of the CYP1B1 protective missense variant.

We thank the reviewer for their positive review of the manuscript.

I have some very minor suggestions and questions:

The overrepresentation of immune genes in cases in the RNA-seq results seems like a very obvious result, and perhaps this is why the authors have not overemphasized findings from the RNA-seq. Figure 1D shows lymphoid cell infiltrate of hair follicles in a patient, but there is no histopathology from a healthy control. Does one expect to see any lymphoid cells at all in a healthy control? If some lymphoid cells are present, the authors may consider using a tool like CYBER SORT to estimate the percentage of immune cells that contribute to the differentially expressed gene list, as a way to confirm the source of the overexpressed genes.

The reviewer makes a salient point that did guide our interpretation of the transcriptomic analysis; given the inflammation at the lesional site, it is not unexpected that we observe apparent upregulation of genes with immune related function. To address the reviewer's specific point, we have undertaken histological analysis on tissue from all 7 healthy controls and these demonstrate no microscopic evidence of a lymphoid cell infiltrate (Supplementary Figure 1).

Further to this, do the authors expect that the under expressed genes in cases result from artefact due to overrepresentation of immune cell transcripts and subsequent underrepresentation of other cell's transcripts?

The reviewer's question relates to the interpretation of the transcriptomic experiment: given that the experiment was undertaken in bulk tissue samples the resulting differential gene expression patterns will both reflect changes in cellular composition as well as specific changes in gene expression within cell populations. The results should therefore be interpreted with this in mind and we have added text to the results and discussion to clarify this point.

It seems that the results presented here implicate CD8 T cell pathology, as did GWAS into alopecia areata. Can the authors comment on the 'repurposing' of treatments effective in AA models and AA clinical trials for FFA?

This is a potentially important implication of the results of the current study and we have added a comment relating to this in the discussion

The implication of the CYP1B1 missense mutation is fascinating. By which mechanism exactly do the authors think that 'reduced clearance of endogenous and exogenous substrates' in the presence of the CYP1B1 risk allele actually contribute to FFA development?

We have amended the statement that the reviewer refers to, to reflect the available evidence and avoid misleading speculation; whilst the variant is demonstrated to reduce the function of CYP1B1 in vitro there is no evidence that it is associated with reduced clearance of endogenous and/or exogenous substrates. Whilst it is appealing to speculate about the interaction of this allele with recently introduced or increasingly prevalent, female-specific exogenous substrates, these hypotheses require careful consideration and experimental investigation.

It is intriguing that metabolites pertinent to the xenobiotic super-pathway were identified in the metabolomics comparison of cases and controls. Approximately 10 of the controls should be heterozygous for the CYP1B1 missense mutation. Is there any evidence for xenobiotic metabolite differences between controls carrying the risk allele and those homozygous for the protective allele? Is there any evidence of xenobiotic metabolite differences in cases when controlling for the CYP1B1 genotype?

Unfortunately, no genotypic data are available for the individuals undergoing metabolomics analysis and the experiment outlined by the reviewer is currently not possible.

If the guidelines allow for more figures, it would be good to include Figures S5 and S8 as main figures, rather than supplementary.

We are limited in the number of figures and tables in the main text. Following comments from Reviewer 1 we have removed Figure S8.

Table S3 should show the allele frequencies of the HLA-B*07:02 alleles in cases and controls. We agree these frequencies are informative and have included them for all HLA alleles (Supplementary Table 4).

Is genotype data available for the metabolite samples?

Unfortunately, no genotypic data is available for the metabolomics cohort.

Typos:

Line 33; “relevance xenobiotic processing” should read “relevance of xenobiotic processing”

Line 299; “been implicated immunity” should read “been implicated in immunity”

Line 307; “oestradiol to and oestrone to” should read “oestradiol and oestrone to”

These typos have all be corrected as indicated.

Reviewer #3 (Remarks to the Author):

In this work, the authors presented a multi-omics study to study frontal fibrosing alopecia, and revealed 4 new disease loci. Here are my major questions:

For either the UK or the Spanish cohort, the cases and controls were typed on different genotyping array. While only probes with same probe design were retained. I am a bit skeptical that potential difference in quality control/genotype calling performed separately in case/control will give rise to false positive results.

The reviewer is right to question the quality control procedures that were undertaken. This was a substantial concern going into the study and we have been cognisant of false positives throughout analysis. There are multiple sources of evidence that give us faith that these results are true positive findings. Firstly, at each associated locus there are >1 independently genotyped SNPs that demonstrate evidence of association consistent with expected LD between them from the 1000 Genomes reference panel. Secondly, cluster plots for each of these SNPs in each of the cohorts have been manually inspected for artefacts and the genotypes are well clustered. Thirdly, the observed allele frequencies of these genotyped variants and imputed variants in these regions are consistent between our control cohorts and other independently ascertained population cohorts (UK Biobank, GNOMAD, TOPMED). Fourthly, we observe associations with the consistent direction and magnitude of effect in both the discovery and replication cohorts.

The authors have to provide more details regarding the plausibility of applying the fine-mapping technique to the relatively low-sample size cohorts

The statistical fine mapping approach utilises an established method defined by Wakefield et al.¹ The approach utilises a simple calculation that includes both the estimated effect size of associated variants as well as the uncertainty in the effect size (with a standard error term), in the situation where there is reduced power to detect association (i.e. smaller sample sizes) the uncertainty in the effect size estimates will be greater, this is accounted for in the fine mapping approach. The only other limitation in applying this approach to studies with relatively small sample sizes is that there is less opportunity to observe a degradation in LD between associated variants in the study sample and therefore the power to discriminate the causal variant is reduced.

The authors should report the lambda GC values in result section. With the given sample size, and using markers with MAF>0.005, it will not be surprising that the lambda GC looks “ok”. I would suggest to compute lambda GC using common variants (e.g. MAF>0.05).

We now present λ_{GC} calculated using variants with MAF >0.05 in the Results section.

In Table 1, the OR from the meta-analysis for HLA-B looks very suspicious, as it is higher than both UK and Spanish cohorts, and it has a surprisingly sig. meta-p-value (when comparing with those from the UK and Spanish cohorts)

We thank the author for highlighting this unfortunate error – this was created in the generation of this table and has now been corrected.

Have the authors look into the overlap between the DEGs from RNA-seq to those loci with at least suggestive evidence of association?

We thank the reviewer for this helpful suggestion. The results of this have now been added to results section of the manuscript.

Are the patients from the metabolomic analysis the same as those from the RNA-seq? In addition, is there any correlation between metabolite values with the gene expression profiles?

The GWAS, RNA-seq and metabolomics cohorts are all independent. Unfortunately, it is therefore not possible to investigate the correlation between gene expression and metabolite profiles.

While I appreciate the authors study the FAA using multi-omics approach, I find the results lack the coherence in presenting how the genetics, transcriptomics, and metabolomics results

We thank the reviewer for their viewpoint: we have re-written parts of the manuscript to improve coherence and ensure clarity in presenting the data.

How did the authors ensure the robustness and generalizability in the RF classification?

Having considered the feedback of reviewer 1, the metabolomic analysis has been simplified and we have removed the RF analysis in the revised manuscript.

References

1. Wakefield, J. Bayes factors for genome-wide association studies: comparison with P-values. *Genet Epidemiol* 2009; **33**: 79-86

Reviewer #1 (Remarks to the Author):

The authors have responded to all of my concerns and suggestions and I think the resulting manuscript is greatly improved.

There are still grammatical errors that need to be addressed. There is a tendency to have lengthy sentences that could be broken into two and there is an over-use of commas throughout the entire manuscript. I've highlighted two examples below, but the manuscript really needs to be carefully reviewed for grammar and removal of unnecessary commas. I recommend publication.

One example of both problems:

“Since FFA was first identified by Kossard in 1994, there has been rapid increase in reported incidence, culminating in intense clinical and public interest in the condition, which is often referred to as a dermatological epidemic, and whether there is a likely environmental trigger.”

Also, I would recommend avoiding a definitive statement that it is the causal allele by interjecting "likely to be" below. Also, another example of grammar/comma in the sentence that follows:

“At 2p21.2, rs1800440, is likely to be the causal variant underlying the association signal with a posterior probability of 0.98. The FFA protective allele is a missense allele (c.1358A>G p.Asn453Ser) in the CYP1B1 gene, introduces a serine residue in the haem binding domain of the enzyme.”

Reviewer #2 (Remarks to the Author):

I am satisfied with all of the authors edits and recommend this manuscript to be accepted.

Reviewer #3 (Remarks to the Author):

The authors have addressed all my previous comments, and the manuscript is greatly improved. I just have one minor suggestion, since the authors claim this to be the first genetic study on FFA, could the authors provide an estimated heritability value using the genome wide genetic data ?

Response to reviewers

Reviewer #1 (Remarks to the Author):

The authors have responded to all of my concerns and suggestions and I think the resulting manuscript is greatly improved. There are still grammatical errors that need to be addressed. There is a tendency to have lengthy sentences that could be broken into two and there is an over-use of commas throughout the entire manuscript. I've highlighted two examples below, but the manuscript really needs to be carefully reviewed for grammar and removal of unnecessary commas. I recommend publication.

One example of both problems:

"Since FFA was first identified by Kossard in 1994, there has been rapid increase in reported incidence, culminating in intense clinical and public interest in the condition, which is often referred to as a dermatological epidemic, and whether there is a likely environmental trigger."

Also, I would recommend avoiding a definitive statement that it is the causal allele by interjecting "likely to be" below. Also, another example of grammar/comma in the sentence that follows: "At 2p21.2, rs1800440, is likely to be the causal variant underlying the association signal with a posterior probability of 0.98. The FFA protective allele is a missense allele (c.1358A>G p.Asn453Ser) in the CYP1B1 gene, introduces a serine residue in the haem binding domain of the enzyme."

We thank the reviewer for their positive appraisal of our revised manuscript. We have now reviewed and edited the entire manuscript accordingly and the highlighted issues have been addressed. Several other sentences have been slightly modified in terms of grammar and use of punctuation (especially commas).

Reviewer #2 (Remarks to the Author):

I am satisfied with all of the authors edits and recommend this manuscript to be accepted.

We thank the reviewer for their positive appraisal of our revised manuscript and for their constructive help with improving our submission.

Reviewer #3 (Remarks to the Author):

The authors have addressed all my previous comments, and the manuscript is greatly improved. I just have one minor suggestion, since the authors claim this to be the first genetic study on FFA, could the authors provide an estimated heritability value using the genome wide genetic data?

We thank the reviewer for their positive appraisal of our revised manuscript. We have now estimated heritability by utilising the established genomic-relatedness-based restricted maximum-likelihood (GREML) approach using the GCTA software and have added these estimates to the results section.

{no additional comments from reviewer #3}